# Evaluation of an online intervention for improving stroke survivors' health-related quality of life: A randomised controlled trial

Ashleigh Guillaumier[1,2]*, Neil J. Spratt[1,2,3], Michael Pollack[1,3], Amanda Baker[1,2], Parker Magin[1], Alyna Turner[1,4], Christopher Oldmeadow[2], Clare Collins[1,2], Robin Callister[1,2], Chris Levi[1,3], Andrew Searles[2], Simon Deeming[2], Brigid Clancy[1], Billie Bonevski[5]

**1** University of Newcastle, Callaghan, New South Wales, Australia, **2** Hunter Medical Research Institute, New Lambton Heights, New South Wales, Australia, **3** Hunter New England Local Health District, New Lambton Heights, New South Wales, Australia, **4** Deakin University, Geelong, Victoria, Australia, **5** Flinders University, Bedford Park, South Australia, Australia

* Ashleigh.Guillaumier@newcastle.edu.au

**Data Availability Statement:** All data files are available from the University of Newcastle NOVA database available at http://hdl.handle.net/1959.13/1431114.

## Abstract

### Background

The aim of this trial was to evaluate the effectiveness of an online health behaviour change intervention—Prevent 2nd Stroke (P2S)—at improving health-related quality of life (HRQoL) amongst stroke survivors at 6 months of follow-up.

### Methods and findings

A prospective, blinded-endpoint randomised controlled trial, with stroke survivors as the unit of randomisation, was conducted between March 2018 and November 2019. Adult stroke survivors between 6 and 36 months post-stroke with capacity to use the intervention (determined by a score of ≥4 on the Modified Rankin Scale) and who had access and willingness to use the internet were recruited via mail-out invitations from 1 national and 1 regional stroke registry. Participants completed baseline (n = 399) and 6-month follow-up (n = 356; 89%) outcome assessments via computer-assisted telephone interviewing (CATI). At baseline the sample had an average age of 66 years (SD 12), and 65% were male. Randomisation occurred at the end of the baseline survey; CATI assessors and independent statisticians were blind to group allocation. The intervention group received remote access for a 12-week period to the online-only P2S program (n = 199; n = 28 lost at follow-up). The control group were emailed and posted a list of internet addresses of generic health websites (n = 200; n = 15 lost at follow-up). The primary outcome was HRQoL as measured by the EuroQol Visual Analogue Scale (EQ-VAS; self-rated global health); the outcome was assessed for differences between treatment groups at follow-up, adjusting for baseline measures. Secondary outcomes were HRQoL as measured by the EQ-5D (descriptive health state), diet quality, physical activity, alcohol consumption, smoking status, mood, physical functioning, and independent living. All outcomes included the variable 'stroke event

**Funding:** This study was funded by a grant from the National Health and Medical Research Council (NHMRC) (APP1125429) awarded to BB, NJS, MP, AB, PM, AT, CO, CC, RC; https://www.nhmrc.gov.au/. AG is supported by a post-doctoral fellowship from the Heart Foundation, award number 101303, https://www.heartfoundation.org.au/. NJS was supported by a co-funded Australian NHMRC/National Heart Foundation Career Development/Future Leader Fellowship [GNT1110629/100827]. AB is supported by a NHMRC Research Fellowship (G1200044). The funders had no role in study design, data collection and analysis, decision to publish, or preparation of the manuscript.

**Competing interests:** The authors have declared that no competing interests exist.

**Abbreviations:** AUDIT-C, Alcohol Use Disorders Identification Test–Consumption; AuSCR, Australian Stroke Clinical Registry; CATI, computer-assisted telephone interviewing; EQ-VAS, EuroQol Visual Analogue Scale; GLTEQ, Godin Leisure-Time Exercise Questionnaire; HRQoL, health-related quality of life; HSRVR, Hunter Stroke Research Volunteer Register; IADL, Instrumental Activities of Daily Living; MAR, missing at random; OR, odds ratio; P2S, Prevent 2nd Stroke; PHQ-4, Patient Health Questionnaire–4 item; TIA, transient ischaemic attack.

(stroke/transient ischaemic attack/other)' as a covariate, and analysis was intention-to-treat. At 6 months, median EQ-VAS HRQoL score was significantly higher in the intervention group than the control group (85 vs 80, difference 5, 95% CI 0.79–9.21, $p = 0.020$). The results were robust to the assumption the data were missing at random; however, the results were not robust to the assumption that the difference in HRQoL between those with complete versus missing data was at least 3 points. Significantly higher proportions of people in the intervention group reported no problems with personal care (OR 2.17, 95% CI 1.05–4.48, $p = 0.0359$) and usual activities (OR 1.66, 95% CI 1.06–2.60, $p = 0.0256$) than in the control group. There were no significant differences between groups on all other secondary outcomes. The main limitation of the study is that the sample comprises mostly 'well' stroke survivors with limited to no disability.

## Conclusions

The P2S online healthy lifestyle program improved stroke survivors' self-reported global ratings of HRQoL (as measured by EQ-VAS) at 6-month follow-up. Online platforms represent a promising tool to engage and support some stroke survivors.

## Trial registration

Australian New Zealand Clinical Trials Registry ACTRN12617001205325.

---

### Author summary

#### Why was this study done?

- Stroke can lead to serious consequences for those who survive in terms of physical and cognitive disability, psychological problems, and lower social participation, affecting quality of life.

- Improving lifestyle and health risk behaviours (including reducing tobacco and alcohol use, increasing physical activity, improving diet quality, and reducing depression and anxiety) has the potential to significantly improve recovery, enhance quality of life and independent living, and lower risk of recurrent stroke.

- The prevalence of health risk factors amongst stroke survivors is high. There is a striking lack of information for stroke survivors and their families about effective lifestyle strategies to help them improve recovery and reduce the risk of recurrent stroke.

#### What did the researchers do and find?

- We developed an online program (Prevent 2nd Stroke) providing easily accessible, interactive, tailored healthy lifestyle and behaviour change information that encouraged users to set goals and monitor progress across 6 core modules: (1) smoking, (2) alcohol, (3) activity, (4) nutrition, (5) feelings and mood, and (6) blood pressure.

- Adult stroke survivors ($n$ = 399) participated in a randomised controlled trial. Participants completed a telephone survey and then were randomised to the online Prevent 2nd Stroke program arm (participants received access to the online program and were encouraged to use it over a 12-week period) or a control arm (participants were sent a list of generic health information websites). All participants completed a follow-up survey 6 months after their first survey.

- Participants who had access to the online program rated their overall health and well-being 5 points higher on average than participants who received a generic list of health behaviour information websites.

### What do these findings mean?

- Online platforms are a viable and impactful model to address the health information needs and behaviour change challenges of stroke survivors.

- Future studies should test program adaptations particularly for those with greater stroke-related disability.

## Introduction

Recurrent stroke is the major contributor to stroke-related disability and costs [1]. Stroke recovery and rehabilitation can continue for years post-stroke, meaning it is critical that stroke survivors are provided with and have access to a range of support options and evidence-based information. Supporting healthy recovery and preventing recurrent stroke can reduce disability and costs and improve quality of life. Despite clinical recommendations and evidence of impact, Australian Stroke Foundation audits [2,3] have found that 40% of patients do not receive information on stroke, lifestyle management, second prevention, and recovery at the time of rehabilitation discharge. As stroke has such a wide-ranging impact on individuals, health-related quality of life (HRQoL) is an important outcome in understanding and improving survival as well as informing secondary prevention, with stroke survivors as the centre of care.

On average, HRQoL after stroke is low compared to general population norms for at least 5 years post-stroke [4]. Even in patients with transient ischaemic attack (TIA) or minor strokes, stroke recurrence is associated with poor HRQoL [5]. Mood improvement, cognitive stimulation, and healthier behaviours such as physical activity are linked with better HRQoL amongst stroke survivors [6]. Health risk factors, particularly physical inactivity and obesity, that persist following a stroke may also contribute to compromised functioning and independent living during recovery [7]. Even survivors who are not physically disabled, or who have 'mild' stroke, experience ongoing 'invisible' changes such as fatigue, memory problems, anxiety, mood disturbance, and depression [8]. Depression and anxiety are modifiable affective states associated with increased post-stroke morbidity and mortality [9], as well as reduced social participation [6]. Depression has also been associated with poor adherence to treatment and healthy behaviours [10]. HRQoL has become a key outcome in stroke research, offering a comprehensive and multidimensional assessment of the impact of stroke, recovery, and social, physical, and psychological health from the patient's perspective.

A landmark study identified 10 risk factors associated with 90% of the risk of stroke, including smoking, poor diet, physical inactivity, alcohol intake, depression, waist-to-hip ratio, diabetes mellitus, and cardiac causes [11]. Addressing behavioural and affective risk factors may not only lower the risk of subsequent stroke but may improve recovery from the first stroke and HRQoL. One modelling study suggests that a strategy combining appropriate medication adherence with diet and physical activity could result in a cumulative relative risk reduction for recurrent stroke of up to 80% [12]. Unfortunately, these modifiable risk factors are rarely addressed effectively after stroke. Australian Stroke Foundation audits show that 28% of stroke patients in acute settings and 35% in rehabilitation settings do not receive risk factor education [2,3].

Providing behavioural intervention, such as brief advice, education and, counselling, to modify patient health risk behaviours is evidence-based best practice [13]. Cited barriers to delivery of such care include practitioners' lack of training, confidence, skills, and time to provide counselling and advice [14]. There is a striking lack of information on effective health behaviour change strategies for stroke survivors for prevention of a second event [15]. Although promising, evidence from secondary prevention trials [16] is sparse and based on shared care and nurse-led programs that are reliant on substantial resources and are costly to the health service.

Online programs are increasingly popular and may reach a higher number of people than face-to-face programs, including people with mobility restrictions. Up to 80% of patients have an interest in supplementing clinician-delivered support with web-delivered information [17]. Reviews have consistently demonstrated the effectiveness of online interventions both with non-clinical populations for health behaviour change [18] and with co-morbid populations for reducing pain, disability, depression, and anxiety [19,20]. Only 3 small pilot trials of online health programs for stroke survivors have been reported, but they show promising results [21–23]. There is a need for well-powered, well-designed trials of the effectiveness of online interventions for improving quality of life and health behaviours for stroke survivors.

The primary aim was to evaluate the effectiveness of an online health behaviour change intervention—Prevent 2nd Stroke (P2S)—at improving HRQoL amongst stroke survivors at 6 months of follow-up. The development of the P2S program [24] and acceptability and feasibility piloting [25] of the program have been reported elsewhere. Secondary aims were to examine the effect of the online P2S program on 4 health behaviours (smoking, alcohol use, fruit and vegetable intake, and moderate physical activity), mental health (depression and anxiety), and self-reported physical functioning and independent living.

## Methods

### Design

A prospective, blinded-endpoint randomised controlled trial with stroke survivors as the unit of randomisation was conducted between March 2018 and November 2019. We have reported our trial according to the Consolidated Standards of Reporting Trials (CONSORT) guidelines (S1 CONSORT Checklist). The full study protocol has been described in detail [26]. Briefly, independent computer-assisted telephone interviewing (CATI) assessors conducted baseline surveys with all participants. The research team mailed correspondence to participants alerting them to group allocation and providing 12-week access to the P2S program for those in the intervention group. Six months following the baseline survey, the blinded CATI assessors conducted follow-up telephone surveys. Participants were considered 'lost to follow-up' if they were unable to be contacted at the 6-month assessment time point. The study received ethics approval from the University of Newcastle Human Research Ethics Committee (H-2017-0051).

## Recruitment

Participants were recruited through 2 sources: (1) the Australian Stroke Clinical Registry (AuSCR) database and (2) the Hunter Stroke Research Volunteer Register (HSRVR). The AuSCR and HSRVR screened their database registrants against the study eligibility criteria and sent invitation packs on behalf of the study team to potentially eligible individuals. Eligible participants were aged 18 years and over, had been admitted to an AuSCR hospital for acute stroke or TIA (indexed stroke event) or were registered with the HSRVR, were between 6 and 36 months post-stroke, were sufficiently fluent in English, and had sufficient facility in internet use via a home device (e.g., phone, computer, tablet device) or were willing to use public internet services (e.g., public library). The invitation packs contained study consent and consent-to-contact forms. If interested, potential participants sent the completed forms directly to the study team. The study team contacted potential participants via telephone to complete eligibility screening, enrolment, and scheduling of the baseline survey. Participants were ineligible to continue in the trial if they experienced disability at a level that would have limited their capacity to use the intervention (determined by a score of ≥4 on the Modified Rankin Scale) [27]. The protocol stated an additional exclusion criterion of 'documented evidence of 2 or more strokes': This exclusion criterion was removed 2 months into the 12-month recruitment period (and all potential participants who had been excluded on this basis prior to the change were recontacted) to ensure that a sufficient sample size for the trial was obtained.

## Randomisation and data collection

Independent CATI assessors, who were unaware of treatment allocation, completed baseline and follow-up surveys via telephone with all participants. A random number generator embedded in the CATI software was used to allocate participants to the intervention or control group after all baseline questions had been answered. Individuals were randomised at a ratio of 1:1 in permuted blocks of randomly varying size, stratified by state (New South Wales/South Australia, Queensland, Western Australia, Victoria/Tasmania), and type of stroke (stroke, TIA, don't know).

To minimise loss to follow-up, participants were asked to provide multiple points of contact (i.e., home phone, mobile, address, email address) and were sent a monthly text message encouraging them to update any out-of-date contact details and providing an approximate date for their 6-month follow-up survey.

## Intervention

Following the baseline survey and randomisation, researchers both emailed and posted letters to notify participants of their group allocation. The intervention period lasted for 12 weeks. During this time all participants received monthly text message reminders to update any outdated contact details; this reminder also provided the approximate date of the follow-up survey. The intervention group received additional fortnightly text message prompts to use the intervention (overlapping P2S program use and contact detail messages were combined, so the intervention group were contacted on a fortnightly basis during the intervention period).

**Intervention group.** The P2S program was developed using behaviour change theory and co-design principles with stroke survivors, and was pilot tested prior to this trial. The development and testing of the intervention have been described in detail elsewhere [24,25]. Briefly, the P2S program is a modularised, tailored program providing evidence-based techniques and information in 6 core modules: (1) blood pressure, (2) smoking, (3) alcohol consumption, (4) physical activity, (5) nutrition, and (6) feelings and mood, as well as a 'my progress' section. Each health risk behaviour module commences with 2–3 brief questions regarding the topic of

interest (e.g., smoking: smoking status, number of cigarettes per day, interest in quitting) in order to provide tailored information to the user. Users were asked to set specific goals within each module (e.g., physical activity: 'I will increase physical activity from 0 to 2 sessions, of 10 mins each, per day') and were provided with information and advice on how to achieve the goal. The advice in each module is tailored to accommodate stroke-related symptoms. Progress against goals was graphed in the 'my progress' section to provide feedback to users on their behaviour change.

**Control group.**   Participants in the control group were both posted and emailed a copy of a letter containing links to internet addresses with readily available, generic online health programs and guidelines designed for the general population (e.g., the Australian Department of Health's How to Quit Smoking website, Australia's Physical Activity and Sedentary Behaviour Guidelines and Eat for Health websites, the National Health and Medical Research Council's Australian Guidelines to Reduce Health Risks from Drinking Alcohol, and the Moodgym website).

## Outcomes

The primary outcome was HRQoL at 6-month follow-up measured using the EuroQol Visual Analogue Scale (EQ-VAS). The EQ-VAS forms one part of the EQ-5D-5L instrument measuring HRQoL [28]. The EQ-VAS records the participant's self-rated health on a scale from 0 ('the worst health you can imagine') to 100 ('the best health you can imagine'). It is a quantitative measure of health outcome that reflects the participant's own judgement. The protocol paper lists both EQ-VAS and EQ-5D (i.e., the entire EQ-5D-5L) as dual primary outcomes; however, upon finalising the statistical analysis plan prior to analysis, the independent statisticians advised nominating a sole primary outcome measure. The EQ-VAS was selected as it was used as the basis for the a priori sample size calculations.

Secondary outcome measures were as follows: additional HRQoL measures using the EQ-5D-5L descriptive system (EQ-5D), which comprises 5 dimensions (mobility, self-care, usual activities, pain/discomfort, anxiety/depression) on a 5-level scale from 'no problems' to 'extreme problems'. Physical functioning and independent living were measured using the Barthel Index [29] and Instrumental Activities of Daily Living (IADL) scales [30]; depression and anxiety, using the Patient Health Questionnaire–4 item (PHQ-4) [31]; diet quality, using the Australian Recommended Food Score (ARFS) [32]; alcohol consumption, using the Alcohol Use Disorders Identification Test–Consumption (AUDIT-C) [33]; physical activity, using the Godin Leisure-Time Exercise Questionnaire (GLTEQ) [34]; and smoking status, using self-reported 7-day point prevalence abstinence [35].

## Sample size

As reported in the protocol, an a priori sample size calculation was conducted. A sample of 160 per treatment arm at follow-up would enable the detection of a 0.25-standard-deviation difference (a 6-point difference) in EQ-VAS (HRQoL) while maintaining a type I error rate of 5% (5% significance threshold) and type II error rate of 20% (80% power). This calculation assumed a correlation between baseline and follow-up HRQoL of 0.6, and a standard deviation of 24 points. Pilot data suggested that 40% of participants recruited at baseline would complete follow-up data [25]. Thus, a target of 530 consenting participants at baseline was set.

## Statistical analysis

Participant characteristics at baseline were compared descriptively. The outcomes were summarised at baseline and follow-up for intervention and control groups. Means, standard

deviations, medians, 25th and 75th percentiles, and minimum and maximum values for continuous variables, and frequencies with percentages of non-missing observations for categorical variables, are provided. Analysis was intention-to-treat.

The outcomes were assessed for differences between treatment groups at follow up. The outcomes EQ-VAS, AFRS, and EQ-5D were adjusted for baseline measures. For the primary outcome, EQ-VAS, median quantile regression was used to assess the difference between treatment groups. This differs from the linear regression specified in the protocol; however, quantile regression, instead of standard normal regression, was chosen due to the EQ-VAS results violating regression assumptions. Ordinary least squares (OLS) regression showed some skewed and heteroskedasticity residuals. Quantile regression is more robust when OLS regression assumptions are violated.

The AFRS and EQ-5D index were assessed using standard linear regression. EQ-5D was assessed with heteroskedasticity-consistent standard errors (sandwich or Huber–White) due to the regression not meeting residual variance assumptions. The Barthel Index, IADL, GLTEQ, and PHQ-4 categories were assessed with ordinal logistic regression. AUDIT-C, PHQ-4 anxiety and depression, and EQ-5D sub-items were assessed with logistic regression.

All outcomes included the randomisation variable 'stroke event' as a covariate. The randomisation variable 'state of residence' was not included due to low numbers in some states. Linear estimates or odds ratios (ORs) along with 95% confidence intervals and $p$-values are provided where appropriate.

Differences between consenters and non-consenters of the AuSCR mail-out were assessed with Fisher's exact tests for categorical variables and $t$ tests for continuous variables.

Missing data were assumed to be missing at random (MAR). Multiple imputation using the chained regression equation method with 50 imputations was used. Model estimates were pooled across imputations using Rubin's method. The pooled estimates are provided, together with 95% confidence intervals and Wald $p$-values. Sensitivity to the MAR assumption was assessed using a tipping point approach, allowing the data to be missing not at random (MNAR). Assuming patients who failed to complete the follow-up questionnaire were likely to have poorer health, a negative shift parameter was added to EQ-VAS during imputation. The shift parameter corresponds to the expected difference in HRQoL between those with and without missing data. A range of shift parameter values were tested, until EQ-VAS no longer showed a statistically significant effect at a $p$-value of 0.05.

Statistical analyses were programmed using SAS version 9.4 (SAS Institute, Cary, North Carolina, US).

## Results

There was a total of 399 consenting participants eligible for randomisation: 199 were randomised to the intervention group, and 200 to the control group. A total of 377 participants were recruited via the AuSCR, while 22 were recruited through the HSRVR. In total, 43 (10.8%) participants were lost to follow-up (including 5 who withdrew) (see Fig 1). No relevant safety events were reported over the course of the trial. Table 1 shows the baseline socio-demographic characteristics of the participants, and Table 2 shows the outcome variable descriptive statistics for baseline and follow-up. Overall, participants were largely an able, independent, non-smoking, healthy eating, and active cohort of stroke survivors, with few indications of anxiety or depression at baseline, although they did report consuming alcohol at high levels. The participant characteristics of the groups were well balanced at baseline. The control group had 15/200 (7.5%) lost to follow-up, while the intervention group had 28/199 (14%). The difference in loss to follow-up between groups was low (6.5%), although statistically significant, $p$ = 0.034.

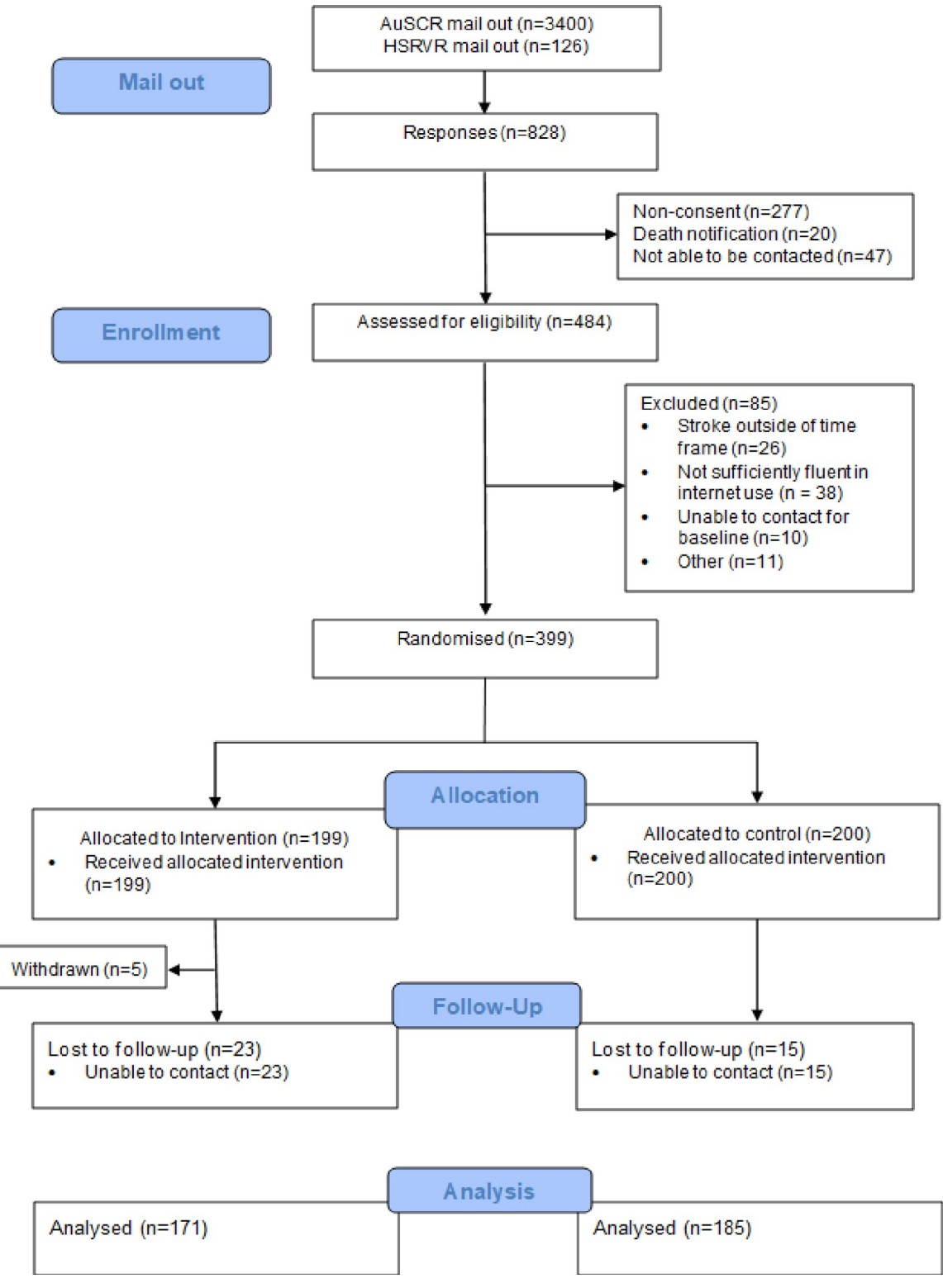

**Fig 1. Study CONSORT flow diagram. AuSCR, Australian Stroke Clinical Registry; HSRVR, Hunter Stroke Research Volunteer Register.**

**Table 1. Demographic characteristics by treatment group at baseline.**

| Variable | Statistic/category | Control (*n* = 200)* | Intervention (*n* = 199)* |
|---|---|---|---|
| Age | *n* | 199 | 199 |
| | mean (SD), years | 68 (12) | 67 (12) |
| | median (minimum, maximum), years | 70 (20, 90) | 68 (20, 93) |
| Sex | Female | 66 (33%) | 73 (37%) |
| | Male | 134 (67%) | 126 (63%) |
| State | New South Wales | 20 (10%) | 21 (11%) |
| | Queensland | 89 (45%) | 88 (44%) |
| | South Australia | 1 (0.5%) | — |
| | Tasmania | 9 (4.5%) | 12 (6.0%) |
| | Victoria | 79 (40%) | 77 (39%) |
| | Western Australia | 2 (1.0%) | 1 (0.5%) |
| Stroke type | Don't know | 8 (4.0%) | 10 (5.0%) |
| | Stroke | 120 (60%) | 119 (60%) |
| | Transient ischaemic attack | 72 (36%) | 70 (35%) |
| Country of birth | Australia | 151 (76%) | 156 (78%) |
| | Other | 48 (24%) | 43 (22%) |
| Indigenous status | Aboriginal and/or Torres Strait Islander | 2 (1.0%) | 1 (0.5%) |
| | Neither | 196 (99%) | 196 (99.5%) |
| Income | ≤AU$399 (US$304) per week | 57 (29%) | 52 (26%) |
| | AU$400–AU$999 (US$305–US$762) per week | 82 (41%) | 75 (38%) |
| | ≥AU$1,000 (US$763) per week | 46 (23%) | 50 (25%) |
| | Don't know/refused | 14 (7%) | 20 (10%) |
| Walk on admission to hospital (time of stroke) | Yes | 102 (51%) | 118 (59%) |
| | No | 85 (43%) | 68 (34%) |
| | Unknown | 13 (6.5%) | 13 (6.5%) |

Data are given as *n* (percent) unless otherwise indicated.

*Not all *n*'s add to 100% due to missing data.

Of the participants allocated to the intervention group, 159/199 (80%) accessed and engaged with the online program. The program was designed so that participants could engage with the content as they chose, and overall module access was as follows: nutrition, 120/159 (75%); blood pressure, 119/159 (75%); physical activity, 104/159 (65%); feelings and mood, 100/159 (63%); and alcohol use, 82/159 (52%). Due to a technical error, the back-end metrics for the smoking module were not produced, although the module itself was active and available to the users.

Table 3 summarises the results of the trial for the primary and secondary outcomes. For the primary outcome, the median HRQoL score (EQ-VAS) was higher in the intervention group than the control group at 6-month follow-up (85 versus 80; difference = 5, 95% CI 0.79–9.21, $p = 0.020$).

The secondary outcome EQ-5D sub-item data at follow-up are presented in Table 4. The majority (>60%) of participants had no problems on any of the items. Table 5 summarises the results of the effects of the trial on the EQ-5D sub-items. While the change in the odds of no problems appears higher in the intervention group for all 5 items, significantly higher proportions of people in the intervention group reported no problems with personal care (159 [93%] versus 159 [86%]; OR 2.17, 95% CI 1.05–4.48, $p = 0.0359$) and usual activities (122 [71%] versus 111 [60%]; OR 1.66, 95% CI 1.06–2.60, $p = 0.0256$) than in the control group.

**Table 2. Outcome variable descriptive statistics for baseline and follow-up time points.**

| Variable | Statistic/category | Baseline | | Follow-up | |
|---|---|---|---|---|---|
| | | Control (n = 200)* | Intervention (n = 199)* | Control (n = 185)* | Intervention (n = 171)* |
| EQ-VAS | n | 199 | 197 | 185 | 171 |
| | mean (SD) | 77.4 (15.9) | 79.5 (15.0) | 78.1 (17.2) | 80.4 (16.8) |
| | median (min, max) | 80 (20, 100) | 80 (20, 100) | 80 (8, 100) | 85.0 (9, 100) |
| | median (Q1, Q3) | 80 (70, 90) | 80 (72, 90) | 80 (70, 90) | 85.0 (75, 90) |
| EQ-5D-5L (UK) | n | 198 | 197 | 185 | 171 |
| | mean (SD) | 0.82 (0.17) | 0.83 (0.18) | 0.82 (0.17) | 0.85 (0.17) |
| | median (min, max) | 0.84 (0.10, 1.00) | 0.84 (−0.06, 1.00) | 0.84 (0.11, 1.00) | 0.88 (−0.06, 1.00) |
| | median (Q1, Q3) | 0.84 (0.73, 1.00) | 0.84 (0.74, 1.00) | 0.84 (0.74, 1.00) | 0.88 (0.77, 1.00) |
| Barthel Index | Moderate (61–90) | 13 (6.5%) | 11 (5.7%) | 20 (11%) | 10 (5.9%) |
| | Slight (91–99) | 41 (21%) | 30 (15%) | 23 (13%) | 29 (17%) |
| | Independent (100) | 145 (73%) | 153 (79%) | 141 (77%) | 130 (77%) |
| IADL | 0–6 | 14 (7.0%) | 17 (8.6%) | 16 (8.6%) | 9 (5.3%) |
| | 7 | 24 (12%) | 15 (7.6%) | 18 (9.7%) | 20 (12%) |
| | 8 | 161 (81%) | 165 (84%) | 151 (82%) | 142 (83%) |
| ARFS (max 73) | n | 198 | 197 | 185 | 170 |
| | mean (SD) | 40.5 (8.8) | 40.1 (9.0) | 40.8 (8.6) | 40.6 (8.1) |
| | median (min, max) | 41.5 (17.0, 59.0) | 42.0 (3.0, 58.0) | 42.0 (18.0, 57.0) | 41.0 (19.0, 59.0) |
| | median (Q1, Q3) | 41.5 (35.0, 47.0) | 42.0 (34.0, 46.0) | 42.0 (36.0, 47.0) | 41.0 (35.0, 46.0) |
| Smoked in the last 7 days | No | 189 (95%) | 191 (97%) | 175 (99%) | 163 (97%) |
| | Yes | 10 (5.0%) | 6 (3.0%) | 2 (1.1%) | 5 (3.0%) |
| Heavy drinking (AUDIT-C) | No | 108 (54%) | 99 (50%) | 100 (54%) | 83 (49%) |
| | Yes | 91 (46%) | 98 (50%) | 85 (46%) | 88 (51%) |
| GLTEQ | Sedentary (0–13) | 64 (32%) | 52 (26%) | 58 (32%) | 40 (24%) |
| | Moderately active (14–23) | 41 (21%) | 60 (30%) | 30 (16%) | 42 (25%) |
| | Active (24+) | 94 (47%) | 85 (43%) | 95 (52%) | 88 (52%) |
| PHQ-4 | None (0–2) | 140 (71%) | 146 (74%) | 141 (77%) | 137 (81%) |
| | Mild (3–5) | 36 (18%) | 34 (17%) | 23 (13%) | 22 (13%) |
| | Moderate (6–8) | 13 (6.6%) | 12 (6.1%) | 12 (6.6%) | 4 (2.4%) |
| | Severe (9–12) | 7 (3.6%) | 4 (2.0%) | 6 (3.3%) | 7 (4.1%) |

Data are given as n (percent) unless otherwise indicated.

*Not all n's add to 100% due to missing data.

ARFS, Australian Recommended Food Score; AUDIT-C, Alcohol Use Disorders Identification Test–Consumption; EQ-VAS, EuroQol Visual Analogue Scale; GLTEQ, Godin Leisure-Time Exercise Questionnaire; IADL, Instrumental Activities of Daily Living; max, maximum; min, minimum; OR, odds ratio; PHQ-4, Patient Health Questionnaire–4 item; Q1, first quartile; Q3, third quartile.

There were no significant differences between groups for any of the other secondary outcomes at follow-up. In both the intervention and control groups, almost all participants were non-smokers (163 [97%] versus 175 [99%]), participants on average had an excellent varied and healthy diet (mean [SD] ARFS = 40.6 [8.1] versus 40.8 [8.6]), and around half were sufficiently active (88 [52%] versus 95 [52%]), although around half of the sample in both groups were drinking alcohol at risky levels (88 [51%] versus 85 [46%]). Most participants in both the intervention and control groups reported no symptoms of depression or anxiety (137 [81%] versus 141 [77%]) and had high levels of physical functioning (130 [77%] versus 141 [77%]) and independence (142 [83%] versus 151 [83%]).

**Table 3. Regression analysis results for primary and secondary outcomes: Complete case and imputation results.**

| Model outcome | Estimate type | Complete case analysis | | | | | Imputed analysis (*n* = 50) | |
|---|---|---|---|---|---|---|---|---|
| | | *N* | Control group: *N* (%) with characteristic | Intervention group: *N* (%) with characteristic | Estimate (95% CI) | *p*-Value | Estimate (95% CI) | *p*-Value |
| Health today (EQ-VAS) | Median difference | 356 | | | 5.00 (0.79, 9.21) | 0.020 | 4.64 (0.12, 9.16) | 0.044 |
| Barthel Index | OR (ordinal) | 353 | | | 1.07 (0.65, 1.76) | 0.786 | 1.07 (0.66, 1.74) | 0.770 |
| Heavy drinking (AUDIT-C) | OR | 355 | 85 (45.9%) | 88 (51.5%) | 1.23 (0.81, 1.87) | 0.323 | 1.29 (0.86, 1.92) | 0.221 |
| PHQ-4 | OR (ordinal) | 352 | | | 0.82 (0.49, 1.36) | 0.437 | 0.79 (0.48, 1.30) | 0.350 |
| PHQ-4 (anxiety) | OR | 351 | 17 (9.34%) | 13 (7.60%) | 0.80 (0.38, 1.68) | 0.548 | 0.78 (0.38, 1.61) | 0.501 |
| PHQ-4 (depression) | OR | 352 | 22 (11.9%) | 17 (10.0%) | 0.82 (0.42, 1.60) | 0.567 | 0.80 (0.42, 1.54) | 0.507 |
| IADL | OR (ordinal) | 356 | | | 1.14 (0.66, 1.97) | 0.641 | 1.10 (0.65, 1.85) | 0.721 |
| GLTEQ | OR (ordinal) | 353 | | | 1.15 (0.77, 1.71) | 0.497 | 1.13 (0.77, 1.67) | 0.539 |
| ARFS | Mean difference | 353 | | | 0.16 (−1.36, 1.69) | 0.834 | 0.21 (−1.37, 1.79) | 0.795 |
| EQ-5D-5L index (UK) | Mean difference | 354 | | | 0.02 (−0.01, 0.05) | 0.135 | 0.02 (−0.01, 0.05) | 0.107 |
| EQ-5D-5L index (Germany) | Mean difference | 354 | | | 0.01 (−0.01, 0.03) | 0.292 | 0.01 (−0.01, 0.03) | 0.233 |

ARFS, Australian Recommended Food Score; AUDIT-C, Alcohol Use Disorders Identification Test–Consumption; EQ-VAS, EuroQol Visual Analogue Scale; GLTEQ, Godin Leisure-Time Exercise Questionnaire; IADL, Instrumental Activities of Daily Living; OR, odds ratio; PHQ-4, Patient Health Questionnaire–4 item.

Multiple imputation with the MAR assumption did not significantly change the results, indicating that missing data (including loss to follow-up), when assumed to be MAR, did not affect the outcomes of the trial. Tipping point analysis showed that the difference in HRQoL score between intervention and control would lose statistical significance at a shift parameter of −3. A difference of 3 between those missing and not missing data is not unrealistic, meaning that the results may not be robust to the MAR assumption.

## Discussion

The results indicate the P2S online healthy lifestyle program improved stroke survivors' self-reported global rating of HRQoL (as measured by EQ-VAS) at 6-month follow-up. Additionally, more stroke survivors in the intervention group reported having no problems with personal care and usual activities compared to those in the control group (as measured by EQ-5D) at 6-month follow-up. There appeared to be no difference between groups at 6-month follow-up for the 4 health risk behaviours (smoking, diet, alcohol, and physical activity), mental health, or physical functioning and independent living.

The stroke survivors who participated in this study were largely independent, high functioning, and healthy. At baseline, individuals in the control and intervention groups rated themselves in top health (i.e., approximately 80 on the EQ-VAS) at around the same levels as in a non-stroke general population [36]. In previous stroke rehabilitation studies, survivors rated their global health on the EQ-VAS on average around 60 [37]. A ceiling effect appears for the EQ-VAS in healthy populations around 90 points, and there is some indication that minimally important differences in stroke populations on the EQ-VAS scale are around 6–8 points. However, this is based on data obtained during the immediate post-stroke recovery period (i.e., up to 3 months post-stroke, when there is a greater capacity to improve) [38]. It could be expected that a sample with on average top health at baseline might prove harder to produce a change within, and therefore it is remarkable to see a significant 5-point increase in

**Table 4. EQ-5D sub-item data at follow-up.**

| Sub-item | Control (n = 185) | Intervention (n = 171) | Total (N = 356) |
|---|---|---|---|
| **Mobility (walking around)** | | | |
| No problems | 113 (61%) | 120 (70%) | 233 (65%) |
| Slight problems | 46 (25%) | 34 (20%) | 80 (22%) |
| Moderate problems | 26 (14%) | 14 (8.2%) | 40 (11%) |
| Severe problems | 0 | 3 (1.8%) | 3 (0.8%) |
| **Personal care (washing/dressing)** | | | |
| No problems | 159 (86%) | 159 (93%) | 318 (89%) |
| Slight problems | 18 (9.7%) | 7 (4.1%) | 25 (7.0%) |
| Moderate problems | 8 (4.3%) | 5 (2.9%) | 13 (3.7%) |
| **Usual activities** | | | |
| No problems | 111 (60%) | 122 (71%) | 233 (65%) |
| Slight problems | 48 (26%) | 33 (19%) | 81 (23%) |
| Moderate problems | 20 (11%) | 13 (7.6%) | 33 (9.3%) |
| Severe problems | 6 (3.2%) | 1 (0.6%) | 7 (2.0%) |
| Unable to do my usual activities | 0 | 2 (1.2%) | 2 (0.6%) |
| **Pain or discomfort** | | | |
| None | 113 (61%) | 113 (66%) | 226 (63%) |
| Slight | 43 (23%) | 34 (20%) | 77 (22%) |
| Moderate | 23 (12%) | 21 (12%) | 44 (12%) |
| Severe | 6 (3.2%) | 3 (1.8%) | 9 (2.5%) |
| **Anxiety or depression** | | | |
| Not anxious or depressed | 127 (69%) | 126 (74%) | 253 (71%) |
| Slightly anxious or depressed | 37 (20%) | 31 (18%) | 68 (19%) |
| Moderately anxious or depressed | 17 (9.2%) | 11 (6.4%) | 28 (7.9%) |
| Severely anxious or depressed | 3 (1.6%) | 0 | 3 (0.8%) |
| Extremely anxious or depressed | 1 (0.5%) | 3 (1.8%) | 4 (1.1%) |

Data are given as n (percent).

intervention EQ-VAS ratings. In any case, there is a need to test this style of intervention in a less well population.

The health risk behaviours assessed in this study indicate that the individuals in the sample were relatively healthy at baseline, which may explain why no changes in these behaviours

**Table 5. Logistic regression results for dichotomised (no problems versus some problems) EQ-5D sub-items.**

| Model outcome | Complete case analysis | | | | Imputed analysis (n = 50) | |
|---|---|---|---|---|---|---|
| | Control group: N (%) with no problems | Intervention group: N (%) with no problems | Odds ratio (95% CI) | p-Value | Odds ratio (95% CI) | p-Value |
| Mobility | 113 (61.1%) | 120 (70.2%) | 1.50 (0.96, 2.34) | 0.0746 | 1.44 (0.93, 2.22) | 0.101 |
| Personal care | 159 (85.9%) | 159 (93.0%) | 2.17 (1.05, 4.48) | 0.0359 | 2.08 (1.03, 4.20) | 0.040 |
| Usual activities | 111 (60.0%) | 122 (71.3%) | 1.66 (1.06, 2.60) | 0.0256 | 1.60 (1.03, 2.48) | 0.038 |
| Pain | 113 (61.1%) | 113 (66.1%) | 1.24 (0.80, 1.92) | 0.3341 | 1.17 (0.76, 1.79) | 0.477 |
| Anxiety/ depression | 127 (68.6%) | 126 (73.7%) | 1.28 (0.81, 2.03) | 0.2942 | 1.27 (0.81, 2.00) | 0.288 |

Data are given as n (percent).

were seen over the intervention period. In our study, the rate of smoking (4%) was lower than the approximately 14% seen in large stroke patient cohort studies [11]. Similarly, there was limited scope for diet improvement as the average diet ratings of the sample were in the 'excellent' category, indicating a nutritious and varied diet aligning with the Australian Dietary Guidelines [32]. Regarding physical activity, half the sample were 'active' and a further quarter 'moderately active' at baseline, and this remained stable over the intervention period. The individuals in the study sample overall were more active than the general population [39]. It may be that diet and exercise were addressed in this sample's post-stroke rehabilitation care, accounting for such high baseline levels. However, given that post-stroke awareness and knowledge of risk factors such as diet and exercise is typically low [40], these findings may also suggest a study population selection bias.

The exception to the largely healthy behaviours reported at baseline among participants was alcohol use. Alcohol use is emerging as a complex area for stroke research. The Mediterranean diet, which allows for moderate daily alcohol consumption, shows a protective cardiovascular effect [41]. While mild-to-moderate alcohol consumption may have a protective cardiovascular effect, this research is not without criticism [42]: Such consumption may present other cancer-related health risks, and hazardous or harmful alcohol consumption does remain a clear risk factor for stroke and stroke recurrence. Future interventions might embed alcohol within diet/nutrition content to ensure brief interventions are achieved.

Rates of anxiety (12%) and depression (12%) at baseline among our relatively 'well' stroke survivor sample were much lower than depression (31%) [43] and anxiety (24%) [44] rates typically reported in stroke survivors. Compared to the general population, the anxiety rate was similar (14%), although the depression rate among our sample (12%) was double the general population rate (6%) [45]. There is a clear relationship between depression and stroke [46], and this can lead to low engagement in protective/preventative health behaviours. The mental health of stroke survivors and its role in prevention of stroke recurrence has been identified as a critical pathway for future research by peak stroke and cardiovascular bodies [47] and requires further attention. Additionally, given the relationship between post-stroke fatigue and depression, where it is common for the conditions to co-occur or for one to be masked by the other [48], it is a limitation of the current study that fatigue was not measured; it should be considered critical to assess fatigue in future studies of this nature.

While the effect of the P2S program is modest overall and may not be robust, an online program can work as a supplementary tool to use at home while still receiving other rehabilitation services. It would be ideal to see a program like P2S offered as part of a toolkit of support options for stroke survivors, and such a program may extend outreach to those who are less well recovered. The Australian Stroke Foundation has developed EnableMe, a similar online platform for resources and information for stroke survivors [49]. The online format is increasingly being found as an acceptable and effective method to deliver health behaviour information to fill the evidence–practice gap in stroke recovery and support.

This trial has a number of strengths and limitations. It was a large trial, recruiting from a national cohort of Australian stroke survivors. As described above, generalisability is limited by the sample comprising mostly 'well' stroke survivors. However, this trial provides a pathway for future testing and development of the program, with the need for program adaptions to be explored, including in the mode of delivery for those with greater stroke-related disability, such as aphasia, or limited internet access or literacy. The sample recruited for this study were willing and able to use online technology. While this does limit immediate generalisability to the wider stroke population, technology use is growing in older populations. Finally, the response rate at follow-up was lower in the intervention group, which may be a reflection of people who didn't engage or felt like they did well enough in the online program not

completing follow-up. However, there was no difference in groups across most outcomes, so this is unlikely. Supplementary analysis on differences between those who completed the study and those who were lost to follow-up found no differences on any demographic variables (see S1 Table), and the outcome variables were included in the imputation analysis presented in the paper, which would account for any differences between those missing and not missing data in the analysis. Imputation under MAR modelling did not produce any different results; however, the robustness of the MAR assumption is in question. If non-completers had poorer health, the intervention would likely produce a non-significant result.

The results of the study show that an online program delivering health behaviour change information improved the self-rated HRQoL of stroke survivors at 6-month follow-up. This indicates that prevention and health risk behaviour change care provision through an online platform is an effective model to engage, support, and improve the lives of stroke survivors.

## Supporting information

**S1 CONSORT Checklist.**
(DOCX)

**S1 Table. Demographic comparison of study completers versus those lost to follow-up.**
(DOCX)

## Author Contributions

**Conceptualization:** Ashleigh Guillaumier, Neil J. Spratt, Michael Pollack, Amanda Baker, Parker Magin, Alyna Turner, Christopher Oldmeadow, Clare Collins, Robin Callister, Chris Levi, Andrew Searles, Simon Deeming, Billie Bonevski.

**Data curation:** Ashleigh Guillaumier, Simon Deeming, Brigid Clancy, Billie Bonevski.

**Formal analysis:** Christopher Oldmeadow.

**Funding acquisition:** Ashleigh Guillaumier, Neil J. Spratt, Michael Pollack, Amanda Baker, Parker Magin, Alyna Turner, Christopher Oldmeadow, Clare Collins, Robin Callister, Chris Levi, Andrew Searles, Billie Bonevski.

**Investigation:** Ashleigh Guillaumier, Neil J. Spratt, Michael Pollack, Amanda Baker, Parker Magin, Alyna Turner, Clare Collins, Robin Callister, Chris Levi, Andrew Searles, Simon Deeming, Brigid Clancy, Billie Bonevski.

**Methodology:** Ashleigh Guillaumier, Neil J. Spratt, Michael Pollack, Amanda Baker, Parker Magin, Alyna Turner, Clare Collins, Robin Callister, Chris Levi, Andrew Searles, Simon Deeming, Billie Bonevski.

**Project administration:** Ashleigh Guillaumier, Brigid Clancy, Billie Bonevski.

**Resources:** Amanda Baker, Clare Collins.

**Supervision:** Ashleigh Guillaumier, Neil J. Spratt, Michael Pollack, Amanda Baker, Parker Magin, Alyna Turner, Christopher Oldmeadow, Clare Collins, Robin Callister, Simon Deeming, Billie Bonevski.

**Writing – original draft:** Ashleigh Guillaumier.

**Writing – review & editing:** Ashleigh Guillaumier, Neil J. Spratt, Michael Pollack, Amanda Baker, Parker Magin, Alyna Turner, Christopher Oldmeadow, Clare Collins, Robin Callister, Chris Levi, Andrew Searles, Simon Deeming, Brigid Clancy, Billie Bonevski.

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
