## [Editor Report · Decision Letter 0]

19 Nov 2021

Dear Dr Guillaumier, 

Thank you for submitting your manuscript entitled "An online intervention for improving stroke survivors’ health related quality of life: a randomised controlled trial" for consideration by PLOS Medicine.

Your manuscript has now been evaluated by the PLOS Medicine editorial staff and I am writing to let you know that we would like to send your submission out for external peer review.

Please re-submit your manuscript within two working days, i.e. by Nov 23 2021 11:59PM.

Kind regards,

Callam Davidson

Associate Editor

PLOS Medicine

---

## [Decision Letter · Decision Letter 1]

10 Jan 2022

Dear Dr. Guillaumier,

Thank you very much for submitting your manuscript "An online intervention for improving stroke survivors’ health related quality of life: a randomised controlled trial" (PMEDICINE-D-21-04741R1) for consideration at PLOS Medicine. 

Your paper was evaluated by an associate editor and discussed among all the editors here. It was also discussed with an academic editor with relevant expertise, and sent to independent reviewers, including a statistical reviewer. The reviews are appended at the bottom of this email.

In light of these reviews, I am afraid that we will not be able to accept the manuscript for publication in the journal in its current form, but we would like to consider a revised version that addresses the reviewers' and editors' comments. Obviously we cannot make any decision about publication until we have seen the revised manuscript and your response, and we plan to seek re-review by one or more of the reviewers. 

We hope to receive your revised manuscript by Jan 31 2022 11:59PM. Please email us (plosmedicine@plos.org) if you have any questions or concerns.

We look forward to receiving your revised manuscript. 

Sincerely,

Callam Davidson, 

Associate Editor

PLOS Medicine

plosmedicine.org

Abstract

Please report your abstract according to CONSORT for abstracts, following the PLOS Medicine abstract structure (Background, Methods and Findings, Conclusions) http://www.consort-statement.org/extensions?ContentWidgetId=562

Abstract Methods and Findings:

* Please include the population and setting and provide a summary of the population demographics.

* Please specify who was blinded to the intervention and control.

* Please state that analysis was intention to treat (in both the abstract and the methods).

* Please provide the number of participants and the number of participants lost to follow up in each group.

* Please include relevant p values when quantifying your results.

* Please include the important dependent variables that are adjusted for in the analyses.

In the Abstract Conclusions, please interpret the study based on the results presented in the abstract, emphasizing what is new without overstating your conclusions.

General

Please remove the table of abbreviations and instead define abbreviations on first use.

Citations should be in square brackets, and preceding punctuation.

Methods and Results

Thank you for completing the CONSORT checklist - please update to use section and paragraph numbers, rather than page numbers (as these are likely to change during the revision process). 

Please define the abbreviations in Figure 1.

Please define "lost to follow-up" as used in this study. Other reasons for exclusion should be defined.

The protocol states that the research team ‘will monitor the ethics and safety of the trial’ – please confirm whether any relevant safety events were reported.

Please provide the actual numbers of events for the outcomes, not just summary statistics or ORs.

Please define the abbreviations in Table 3.

Other

All content on page 16 (author contributions, funding, and disclosures) can be removed as this information will be captured as metadata based on your responses to the submission form.

Please only use et al. after listing the first six authors in your references and use the journal name abbreviations found in the National Center for Biotechnology Information (NCBI) databases. See here for further information: https://journals.plos.org/plosmedicine/s/submission-guidelines#loc-references

Comments from the reviewers:

Reviewer #1: Statistical review

This paper reports a randomised controlled trial comparing an online intervention for stroke survivors. The authors demonstrate that this provides significant better primary outcome compared to a control group. 

I had some comments on the paper, split into major and minor:

Major comments

1. Outcomes, page 8: I found it a bit confusing about what the primary outcome was - the EQ VAS is mentioned in the protocol paper although not clearly as the primary outcome; it's not mentioned in the ANZCTR (just EQ-5D-5L). I'm not an expert in the EQ-5D measurement which might explain my confusion, but I would welcome clarification on whether the specific primary outcome used in this paper was unambiguously pre-defined. This is especially true as EQ VAS results are significant but EQ-5D-5L results are not (although not far off).

2. Statistical analysis/Results: I note there was quite strong difference in missing data rates between arms. Since it's possible that the missing data may be missing not at random (it is plausible that individuals with worse outcomes were more likely to be lost to follow-up), are the authors concerned that this may explain the significant difference in results? This is especially true as the intervention did not significantly improve the behaviours that would be the mechanism to improving QoL. It may be useful to demonstrate that a large difference in missing outcomes (compared to computed outcomes assuming MAR) would be required to explain the significant results (a tipping point sensitivity analysis

Minor comments

3. Abstract: It is not clear if the reported median and 95% CI is the improvement in the intervention group or the difference between arms. I note later that it is the difference - I would recommend making this clearer. Please add the p-value for significance.

4. Abstract: Assuming the secondary outcomes listed are all of the secondary outcomes, I'd recommend changing 'included' to 'were'. 

5. Abstract: As this was a prospectively powered trial it is appropriate to provide p-values in addition to the confidence intervals.

6. Statistical analysis, page 9: I understand the reason for using quantile regression instead of the analysis specified in the protocol, but I would recommend that it is highlighted as a difference to the protocol.

7. Page 11 - as per comment 3, please make clearer this is the between group difference and add the p-value.

James Wason

Reviewer #2: 

Important and often understudied work on improving HRQoL using an online behavioral intervention in stroke survivors.

Incomplete information provided for funders (e.g., websites). In the abstract and introduction, it would help to define or make clear the difference between the primary and secondary HRQoL outcomes. Methods: How was “sufficient facility in Internet use” determined? The inclusion criteria state acute stroke or TIA while randomization was stratified by type of stroke defined as stroke TIA or other. Please clarify what is meant by “other.” Please clarify what additional “prompts” were used in the intervention group and frequency. Was “time since stroke” considered during data analysis? Were data collected on intervention usability (e.g., number of times accessed the intervention and what resources or modules were accessed, perhaps some more than others)? Discussion: Any differences noted between those that completed the study and those that were lost to follow-up? Perhaps higher rates of depression and anxiety in those lost to follow-up which was significant in the intervention arm. Same question of the 20% in the intervention arm that did not access or engage in the online program during the intervention time frame. Would also help to understand the reasons why these individuals did not access the program (e.g., technical difficulties etc.). 

Reviewer #3: Abstract

First sentence implies this is a study about secondary prevention-which is not consistent with the second sentence about quality of life. 

Surveys are mentioned in the abstract-presumably these were the outcome questionnaires? If so can they clarify this please?

Primary outcome-should this be the 'change in VAS between baseline and 6 months' rather than the VAS? 

Main paper

The rationale for an intervention preventing stroke recurrence having an effect on quality of life (as the primary outcome) needs to be made more clearly.

What was the rationale providing a list of internet links in the control group-was this an 'active control' or 'usual care'. 

Power calculations: 6 point difference in EQ-VAS: why was this chosen and is it of any clinical relevance? 

10.8% were lost to follow-up. The authors could describe how they attempted to minimise loss to follow-up, as this level of loss has the potential to introduce bias into the results. 

Physical activity was measured by self-report-to what extent was this valid? Did the authors consider directly measured activity, and if so, why was self-report chosen?

Fatigue affects around 50% of stroke survivors and can have a major impact on quality of life. Why was fatigue not measured as an outcome? Is it plausible that an on-line programme like this could influence fatigue through a range of different mechanisms, and this might improve HRQUOL?

---

## [Decision Letter · Decision Letter 2]

14 Mar 2022

Dear Dr. Guillaumier,

Thank you very much for re-submitting your manuscript "An online intervention for improving stroke survivors’ health related quality of life: a randomised controlled trial" (PMEDICINE-D-21-04741R2) for review by PLOS Medicine.

I have discussed the paper with my colleagues and the academic editor and it was also seen again by one reviewer. I am pleased to say that provided the remaining editorial and production issues are dealt with we are planning to accept the paper for publication in the journal.

[LINK]

We look forward to receiving the revised manuscript by Mar 21 2022 11:59PM.   

Sincerely,

Callam Davidson, 

Associate Editor 

PLOS Medicine

plosmedicine.org

Requests from Editors:

Please update your title to “Evaluation of an online intervention for improving stroke survivors’ health related quality of life: a randomised controlled trial”.

Please cite your CONSORT checklist in the Methods, with a statement similar to ‘We reported our trial according to the Consolidated Standards of Reporting Trials (CONSORT) guidelines (S1 CONSORT Checklist)’.

The term gender is used in Table 1 and Supplementary Table 1. Please consult http://www.who.int/gender/whatisgender/en/ and confirm that gender is correct term in this context (or whether sex would be more appropriate).

In the references, Journal name abbreviations should be those found in the National Center for Biotechnology Information (NCBI) databases. 

References 5, 11, and 40-42: Please update the DOI to remove https://doi.org/ (such that the format is consistent with e.g. reference 8).

Please add the date of citation to reference 49. 

Comments from Reviewers:

Reviewer #1: Thank you to the authors for revising their paper on the basis of my previous comments. My major concerns have been addressed well and I have no further issues to raise.

[LINK]

---

## [Editor Report · Decision Letter 3]

16 Mar 2022

Dear Dr Guillaumier, 

On behalf of my colleagues and the Academic Editor, Dr Joshua Willey, I am pleased to inform you that we have agreed to publish your manuscript "Evaluation of an online intervention for improving stroke survivors’ health related quality of life: a randomised controlled trial" (PMEDICINE-D-21-04741R3) in PLOS Medicine.

PRESS

Sincerely, 

Callam Davidson 

Associate Editor 

PLOS Medicine